# Mesenchymal Stem Cell-Derived Exosomes Loaded with Selenium or Nano Selenium as a Novel Therapeutic Paradigm for Streptozotocin-Induced Type 1 Diabetes in Rats

**DOI:** 10.3390/biology13040253

**Published:** 2024-04-11

**Authors:** Dlovan Y. Khalil, Ridah H. Hussein, Wafaa M. El-Kholy

**Affiliations:** 1Virology Department, Central Health Laboratory, Ministry of Health, Sulaymaniyah 46012, Iraq; 2Department of Biology, College of Science, Slaimani University, Sulaymaniyah 46001, Iraq; ridha.hussein@univsul.edu.iq; 3Zoology Department, Faculty of Science, Mansoura University, Mansoura P.O. Box 11432, Egypt; drwafaa54@mans.edu.eg

**Keywords:** apoptosis, diabetes mellitus, exosomes, inflammation, stem cells, streptozotocin (STZ)

## Abstract

**Simple Summary:**

Type 1 diabetes mellitus is a metabolic disorder characterized by hyperglycemia due to insulin insufficiency as a consequence of the pancreatic β-cells’ auto-immune attack. A potential novel remedy, counting on using mesenchymal stem cell (MSC)-derived exosomes loaded with an antioxidant known for its insulin-mimetic effect like selenium could be useful to address this condition and its related complications. The results indicated a hypoglycemic and antiapoptotic superiority for nano selenium-loaded exosomes rather than elemental selenium.

**Abstract:**

Type 1 diabetes mellitus (T1DM) is a metabolic disorder characterized by hyperglycemia due to insulin insufficiency as a consequence of the pancreatic β-cells’ auto-immune attack. Nowadays, the application of mesenchymal stem cell-derived exosomes (MSCs-Exs) as the main cell-free therapy for diabetes treatment is becoming more and more extensive. In non-autologous therapy, researchers are moving towards a new strategy based on loading MSC-Exs with certain drugs, aimed at maintaining and maximizing the function of exosomes at the function site and enhancing their efficiency and safety. This study aims to explore and compare the therapeutic potentialities of mesenchymal stem cell-derived exosomes (MSCs-Exs) loaded with either selenium (Se) or nano selenium (NSe), a natural antioxidant micronutrient, in the management of T1DM in rats. In our 4-week experiment, six rat groups were included, namely, control, Ex+Se, Ex+NSe, STZ-diabetic (D), D+ Ex+Se, and D+Ex+NSe groups. Both diabetic-treated groups showed marked pancreatic regenerative antioxidant, immunomodulatory, anti-inflammatory, and anti-apoptotic capacities, with the D+Ex+NSe injection showing superiority in managing diabetes hazards, as evidenced by various biochemical and histological assessments.

## 1. Introduction

Type 1 diabetes mellitus (T1DM) is a metabolic disorder characterized by hyperglycemia due to insulin insufficiency as a consequence of the pancreatic β-cells’ auto-immune attack [1,2]. The most common therapeutic strategies were and still are exogenous insulin injection and oral hypoglycemics, which only manage hyperglycemia temporarily but cannot cure diabetic complications [3]. Therefore, a potential novel remedy, counting on using mesenchymal stem cells (MSCs), has emerged for DM treatment, because of their immunomodulatory capability. However, MSC-based therapy highlighted various safety concerns, including their undesired differentiation and the possibility of malignant transformation [4,5]. Of note, the therapeutic effects of MSCs are exerted mainly via paracrine signaling, by secreting various extracellular vesicles (EVs), including microvesicles, apoptotic bodies, and exosomes [3,6].

Exosomes are nano-sized EVs (30–200 nm) that exert their biological effects through the release of many soluble factors, lipids, and RNA, either in a paracrine or endocrine manner [7]. Recently, MSC-derived exosomes (MSC-Exs) have represented a promising cell-free therapy in regenerative medicine [5], where they have been proven to be equally effective [8,9] or even superior [5,6] in the alleviation of T1DM and its complications, compared with their parental MSCs. This excellence may be attributed to their easier quantification, bioactivity maintenance during transportation and storage, intercellular communication roles, and multiple bioactivities, besides the suggestion of a safer therapeutic protocol, because of the lack of direct tumorigenicity and posing of fewer membrane-bound proteins [5,10]. However, exosomes can be developed to perform crucial therapeutic functions by being employed as natural carriers to deliver certain drugs with diverse therapeutic effects, for them to reach full clinical potential [11,12].

Selenium (Se) is an essential micronutrient, that could be administrated as a dietary supplement and plays important roles in normal thyroid function, DNA synthesis, redox regulation, and fertility in humans [13]. In addition, Se was suggested for T1DM management, because of its confirmed strong antidiabetic and insulin-mimetic properties [14,15]. It was reported to protect β-cells from inflammation and oxidative damage in the body, by being involved in the synthesis of 35 different selenoproteins, including glutathione peroxidase (GPx) [13], which can regulate proper insulin-signaling cascade and inhibit β-cells apoptosis [16]. In this line, the treatment of diabetic animals with Se was reported to increase the viability of pancreatic β-cells and insulin secretion, by inhibiting pancreatic apoptosis and necrosis and upregulating the expression of the insulin gene [17,18]. However, Se has a narrow therapeutic window and the toxicity margins are very delicate, whereas the nanoparticles of Se (SeNPs) possess remarkably reduced toxicity [19]. Nanosized Se (NSe) was suggested as a superior effective molecular compound, with higher antidiabetic, antioxidant, and anti-apoptotic activities compared to ordinary Se [20,21], which could be attributed to its improved bio-activity, bioavailability, cellular uptake, delivery, and very limited toxicity [22].

Hence, this study aims to explore and compare the anti-apoptotic and regenerative capabilities of MSCs-EXs loaded with either Se or NSe particles to alleviate hyperglycemia and restore normal insulin secretion from the pancreas of T1DM rats.

## 2. Materials and Methods

### 2.1. Chemicals

Streptozotocin (STZ), selenium (Se), and nano selenium (NSe) were purchased from Sigma Aldrich Co. (St. Louis, MO, USA), while exosomes (500 μg/mL) were purchased from NAWAH Co. (El Mokattam, Egypt).

### 2.2. AD-MSCs-Exs Preparation and Characterization

The exosomes derived from mesenchymal stem cell stock (500 μg/mL) were sonicated in phosphate-buffered saline (Pbs, pH 7.4) for 10 min, to ensure their complete solubilization. The Ex morphological identity was confirmed using a transmission electron microscope (TEM, Zeiss, LEO 906E, Jena, Germany). A FACS Caliber flow cytometer (Becton Dickinson, Franklin Lakes, NJ, USA) was applied for the characterization of specific surface markers (CD9, CD63, and CD81) (Sigma Aldrich, USA), by using specific rats’ antibodies purchased from Sigma Aldrish, USA [4,6].

### 2.3. Nano-Selenium (NSe) Preparation and Characterization

NSe particles (5–50 nm) were purchased from Sigma Aldrich (USA), separated by centrifugation, and dispersed in an aqueous medium by means of sonication. Meanwhile, TEM (Zeiss, LEO 906E, Germany) was used to examine the NSe particles’ shape and size [21].

### 2.4. Loading of EXs with Either Se or NSe

Se or NSe particle incorporation into exosomes was performed using a sonication-aided exosome drug-loading technique according to Haney et al. [23], with modifications. Exosomes were dispersed in 0.1 mg/mL PBS. Either 20 mL of Se solution in DMSO (200 ppm) or 20 mL of NSe in PBS were added to 20 mL of exosome solution forming a Drug-EXs mixture, which was then sonicated using a probe sonicator (750 v, 20% power, 6 cycles by 4 s pulse/2 s pause (total time 24 s), and cooled down for 2 min on ice, before being further sonicated. These steps were repeated 3 times. The drug-loaded exosomes were enclosed in a cellulose dialysis sac (12–14 KDa) for purification. The dialysis bag was sealed properly both from top and bottom and inserted into 7.5 mL of PBS 7.4 at 50 r.p.m. using a shaking incubator at room temperature. After 1 h, the media were collected and replaced with new 7.5 mL of PBS for another 30 min. On the other hand, the same steps were performed to create unloaded exosomes but without adding active ingredients and using PBS only. The obtained sample stocks were divided into insulin syringes (each of 0.5 mL), covered with foil, and stored at −20 °C until use.

### 2.5. Induction of Diabetes

Rats were fasted overnight and then received an intraperitoneal (IP) single dose of streptozotocin (STZ, 60 mg/kg bw) solution dissolved in cold sodium citrate buffer (0.1 M, pH 4.6). After 3 days, the tail vein fasting blood glucose (FBG) level was assessed using an ACCU–CHEK Go glucometer (Roche Co, Mannheim, Germany). Only rats with FBG over 200 mg/dL were considered diabetics [24].

### 2.6. Experimental Design

The 4-week experimental design was approved by the Animal Ethics Committee of the Faculty of Science, Mansoura University, Mansoura, Egypt [MU-ACUC (SC.PhD.23.03.6)]. Sixty-four male albino Wistar rats (180–210 g) were housed in plastic cages with water and chow free access. Animals were split into 6 groups each of 8 rats, as follows:A.Control groups
**1.** **Normal group:** Rats received an IP single dose of citrate buffer (2 mL/kg, pH 4.6).**2.** **Ex group: Rats** received an IV single dose of 0.5 mL unloaded EXs.**3.** **Ex+Se group: Rats** received an IV single dose of 0.5 mL EXs loaded with Se.**4.** **Ex+NSe group:** Rats received an IV single dose of 0.5 mL EXs loaded with NSe.B.Diabetic groups
**5.** **Diabetic (D) untreated group:** Diabetic rats received an IP single dose of STZ (60 mg/kg).**6.** **Ex-treated group:** Diabetic rats received an IV single dose of 0.5 mL unloaded EXs.**7.** **Diabetic Ex+Se-treated group:** Diabetic rats received an IV single dose of 0.5 mL EXs loaded with Se.**8.** **Diabetic Ex+NSe-treated group:** Diabetic rats received an IV single dose of 0.5 mL EXs loaded with NSe.

### 2.7. Sample Collection

Diethyl ether was used to anesthetize overnight fasted rats before dissecting them. A cardiac puncture was performed to collect blood, where a few droplets, for the assessment of glycosylated hemoglobin (HbA1c), were placed in heparinized tubes. Blood samples were centrifuged (1000× *g* for 15 min) to obtain plasma, and sera were carefully separated and stored at −20 °C. On the other hand, pancreas specimens were harvested and divided into three appropriate parts; one part for forming 10% (*w*/*v*) homogenate (kept at −20 °C), another fresh part for flow cytometric analysis, and the third part for histopathological examination (stored in 10% formalin solution).

### 2.8. Histochemical Examination

Pancreas samples were cut (1 cm thickness), fixed in 10% neutral buffer formalin, and embedded in paraffin wax to form sample blocks. A microtome was used to cut the wax blocks into 3–5 µm thickness sections, which were later mounted on labeled glass slides. A Leica Auto Stainer was then used to stain the slides with Hematoxylin and Eosin. Afterward, a light microscope was used to examine the pancreatic samples [25].

### 2.9. Biochemical Assays

Bio Vision Company, Milpitas, CA, USA, ELISA kits were used to detect serum insulin and C-peptide levels. The estimation of serum glucose, glycosylated hemoglobin (HbA1c), and amylase levels was performed using SPINREACT diagnostics kits in Spain. The pancreatic content of some antioxidants markers, including catalase (CAT), superoxide dismutase (SOD), glutathione (GSH), and glutathione peroxidase (GPx), in addition to some oxidative stress markers, including nitric oxide (NO), malondialdehyde (MDA), and hydrogen peroxide (H_2_O_2_), were all assessed using diagnostic kits obtained from Bio Diagnostic Company, Egypt.

### 2.10. Flow Cytometric Analysis

Antibody kits obtained from Sigma Aldrich Company (USA) were used to determine various pancreatic inflammatory indicators % (IL-6, TGF-β, and TNF-α), apoptotic (annexin V, P53, caspase-3) and anti-apoptotic (Bcl-2) incidences, and various cell cycle phases, in addition to AD-MSCs-EXs surface markers (CD9, CD63, and CD81), according to the method of Tribukait et al. [26]. Briefly, a fresh tissue specimen was dissolved in 250 mL of distilled water and centrifuged at 1800× *g* rpm for 10 min. the supernatant was aspirated, and cells were fixed in ice-cold 96–100% ethanol. The cells were stained by linking the fluorochrome directly to the primary antibody. Then, the cells were washed with 2 mL of PBS/BSA and centrifuged, where the supernatant was discarded and the cells were resuspended in 0.2 mL of PBS/BSA. Finally, data were acquired by means of flow cytometry using a FACS Caliber flow cytometer (Becton Dickinson, USA) [27].

### 2.11. Statistical Analysis

The obtained data were evaluated using a Statistical Package for the Social Sciences (SPSS/21 software version) for Windows, by applying ANOVA followed by post hoc Tukey multiple range tests. All results were expressed as the mean ± standard error (SEM), where *n* = 8. The level of significance was set at *p* ≤ 0.05 for all statistical tests.

## 3. Results

TEM for the observation of exosome morphology showed circular and intact vesicles with a size range of 100–150 nm (Figure 1a). The flow cytometric analysis showed markedly high CD9 (96.2%), CD63 (99.3%), and CD81 (97.0%) expression, confirming the AD-MSCs-EXs phenotype identity (Figure 1b).

The TEM examination of NSe morphology (Figure 2) indicated a clear, spherical shape, with a size range of 5–50 nm.

A marked elevation in serum levels of both glucose and HbA1c was detected in diabetic rats, as shown in Figure 3A–D, while insulin and C-peptide levels were significantly decreased, compared to control rats. However, a noticeable enhancement was recorded in all diabetic-treated groups, relative to untreated diabetics, with an obvious NSe-loaded Ex hypoglycemic superiority relative to Exs loaded with ordinary Se.

The diabetic rats’ results represented in Figure 4 highlight marked elevations in pancreatic levels of oxidative stress markers (MDA, H_2_O_2_, and NO), relative to the control rats. On the other hand, all treated rat groups reflected a significant oxidative marker decrease, with a marked Ex+NSe treatment excellence.

In comparison with the control rats, the Figure 5 data demonstrate a marked antioxidant content decline in the pancreas of diabetic rats. However, the injection of diabetic rats with either Exs or Exs+Se or Exs+NSe showed critical antioxidant content upregulation compared to the untreated diabetic group. Nevertheless, the Ex+NSe therapy revealed an excellent antioxidant potential compared to the other treatment.

Compared to control rats, the Figure 6 data clearly show a marked progress in the pancreatic inflammatory status of diabetic rats, which was significantly suppressed following the injection with unloaded Exs or Se- and NSe-loaded Exs. Notably, the NSe-loaded Exs revealed the highest anti-inflammatory efficacy.

Relative to control, substantial progress in the apoptotic status of diabetic rats’ pancreas was suggested, as seen in Figure 7, by the upregulation of marked apoptotic markers (P53, Bax, and caspase-3%), coupled with the noted anti-apoptotic Bcl-2% diminishing. In contrast, all therapy protocols revealed a tremendous pancreatic anti-apoptotic effect compared to the untreated diabetic rats, with a superior efficacy for the Ex+NSe protocol.

Figure 8 and Figure 9 showed various pancreatic annexin V % indications in different rat groups. Untreated diabetic rats exhibited a marked decline in viable cell % coupled with significant elevations in apoptotic and necrotic cell %, compared to control. Following different protocols of treatment, diabetic rats showed a notable increase in viable cell %, with a marked decrease in both apoptotic and necrotic cell %. However, the NSe-loaded Ex protocol reflected the greatest improvement in pancreatic cell status.

The Figure 10 and Figure 11 data summarize different pancreatic cell cycle phases in different rat groups. Untreated diabetic rats showed a significant downregulation in the G0/G1 cell %, which reflects the pancreatic viable cell %, compared to control rats. On the contrary, relative to untreated diabetic rats, the Ex, Ex+Se, or Ex+NSe injection to diabetic rats exhibited a marked upregulation in the pancreatic viable cell %. However, the Ex+NSe injection recorded a superior enhancement compared to Ex+Se-treated diabetic rats.

Figure 12a–d demonstrate a representative image for pancreatic tissues of different normal rat groups ***(C***, ***Ex***, ***Ex+Se***, and ***Ex+NSe***, respectively), which showed a standard architecture without any notable pathological changes, where prominent nuclei and clear borders of Langerhans’ islets, with intact construction separated by a normal connective tissue septum, were illustrated, indicating the safe usage of EXs loaded with either Se or Nse. In contrast to the previous groups, the pancreatic tissues of diabetic-untreated rats ***(D)*** represented in Figure 12e clarify the destructive effect of the STZ injection, confirmed by the marked islets’ mass size shrinkage, pyknotic nuclei, and congested blood vessels, besides the limited islet vacuolation. However, Figure 12f–h show that diabetic animals treated with unloaded Exs or Exs loaded with either Se ***(D+Exs+Se)*** or Nse ***(D+Exs+Nse)*** indicated a notable variable degree of amelioration in the islets’ architecture, relative to the untreated diabetic group ***(D)***, represented by the obvious restoration of the normal islets’ mass size and the rare occurrence of either congested blood vessels or apoptotic nuclei. Notably, diabetic rats injected with ***NSe-loaded Exs*** displayed superior pancreatic architecture amelioration, compared to other treatment protocols.

## 4. Discussion

Since the principle mechanism of DM induction relies on the huge induction of oxidative stress in the pancreas, administering hypoglycemic agents with known antioxidant potency could effectively restore normal glycemic control and counteract the resulting diabetic hazards in treated subjects. Hence, we suggested a somewhat complicated mechanism of treatment involving the loading of mesenchymal stem cell-derived exosomes (MSC-Exs) with either selenium (Se) or nano selenium (NSe). Our novel protocol greatly upregulated the pancreatic antioxidant contents of diabetic rats, with a direct subsequent suppression of the oxidative stress status. Such events were strongly suggested to decrease the pancreatic inflammatory progress, leading to a pancreatic apoptosis marked arrest, coupled with a significant elevation in the number of viable pancreatic cells, resulting in the restoration of normal insulin secretion and blood glucose management. Similar our results, Sun et al. [3] and He et al. [5] found that MSCs-Exs effectively decreased FBG, increased insulin release, and alleviated hyperglycemia in STZ-treated diabetic rats [28,29] as well as STZ-treated diabetic mice [30], likely via restoring both normal β-islet structure and function, inhibiting β-cell apoptosis and promoting their proliferation, promoting glycolysis/glycogen synthesis, and inhibiting gluconeogenesis.

Interestingly, co-culturing pancreatic β-islets with MSCs-Exs greatly promotes the transplanted islets’ vascularization, angiogenesis, and survival after engraftment, and reduces their apoptosis and auto-immunity, evidenced by the downregulation of some apoptotic genes (Caspase-3, P53, and BAX), while upregulating the anti-apoptotic Bcl-2 gene expression [31,32,33]. Some earlier studies reported that MSCs-Exs can slow or even suppress the progression of T1DM patients by inhibiting the autoimmune targeting of pancreatic β-islets, by promoting autoreactive T-cell apoptosis and regulatory T-cell proliferation; this leads to enhancing the islets’ regeneration, angiogenesis, and survival, resulting in a rise in insulin secretion and the management of hyperglycemia and its complications [7,33,34]. In addition, there is even a possibility that the reparative properties of MSCs-Exs could cause a functional recovery of the already-existing islets, making its injection safer and potentially equally effective or even superior to the MSCs themselves in T1DM therapy and its complications [3,33,35,36].

On the other hand, selenium (Se)’s antidiabetic action appears to result from its behavior as an insulin-mimetic and as an antioxidant nutrient, since both insulin signaling and secretion are associated with the cellular redox state [14,37]. The administration of sodium selenite to diabetic rats for 4 weeks (5 ppm in drinking water), according to Barakat et al. [17,38], or for 2 weeks (0.173 mg/kg, IP injection), according to Iukuza et al. [39], was found to favor a marked FBG and HbA1c decline and the restoration of normal β-cell architecture, likely by suppressing the synthesis of hepatic glucose, promoting glucose uptake by peripheral tissues and glycogen synthesis. Interestingly, the findings of Harmon et al. [40], Lei and Vatamaniuk [41], and Steinbrenner et al. [42] were suggestive of the possible role of Se in the regulation of the redox state of pancreatic β-cells, by increasing the synthesis of other antioxidant selenoproteins including GPx, and participating in the protection of β-islets’ cell membranes from oxidative damage, with a marked restoration of the normal β-cell mass and insulin release. In addition, Labunskyy et al. [43] reported that Se supplementation increased other major antioxidant contents (GSH, SOD, and CAT). However, other mechanisms were suggested to be partly responsible for Se hypoglycemic action, including the renal glucose excretion acceleration and the suppression of intestinal glucose transport [14]. Regarding nano selenium (NSe), Al-Quraishy et al. [44] reported that the treatment of STZ-diabetic rats with NSe (0.1 mg/kg orally for 28 days) greatly alleviated hyperglycemia, possibly by eliciting insulin-mimetic activity, evidenced by the significant decline in FBG levels coupled with a marked elevation in insulin secretion and liver glycogen content, through enhancing the activity of glucose-6-phosphate dehydrogenase while reversing the abnormal activity of some gluconeogenic and glycolytic liver enzymes.

## 5. Conclusions

The injection of MSCs-EXs either unloaded or loaded with Se or NSe resulted in a marked blood glucose management, in rats with T1DM. However, the injection of NSe-loaded MSCs-Exs exhibited a marked hypoglycemic superiority, with notable pancreatic immunomodulatory, anti-inflammatory, anti-apoptotic, antioxidant, and regenerative potentialities. Such behavior is explained, at least in part, by NSe’s recorded antioxidant efficacy in scavenging the pancreatic free radicals, in addition to this metal’s well-known insulin-mimetic effect. However, some limitations should be addressed in future work including (1) the addition of a positive treatment control with a standard anti-diabetic drug such as insulin or metformin, (2) knowing the actual quantity of NSe that was successfully entrapped/loaded into the exosomes, (3) a robust physicochemical characterization, and, finally, (4) ascertaining the therapeutic limits of safety, therapeutic efficacy, and toxicity of the drug, since only ‘one volume, 0.5 mL’ was tested.

## Figures and Tables

**Figure 1 biology-13-00253-f001:**
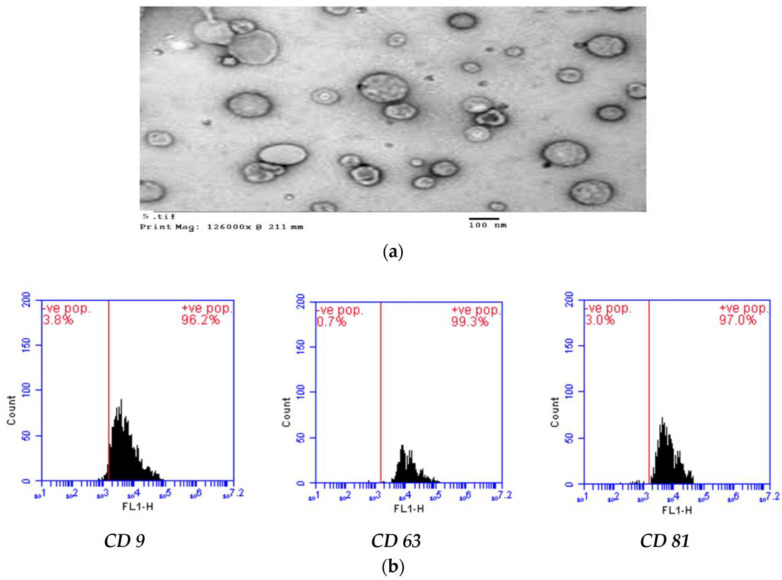
(**a**) Exs morphology. (**b**) MSCs-Exs surface marker flow cytometric analysis.

**Figure 2 biology-13-00253-f002:**
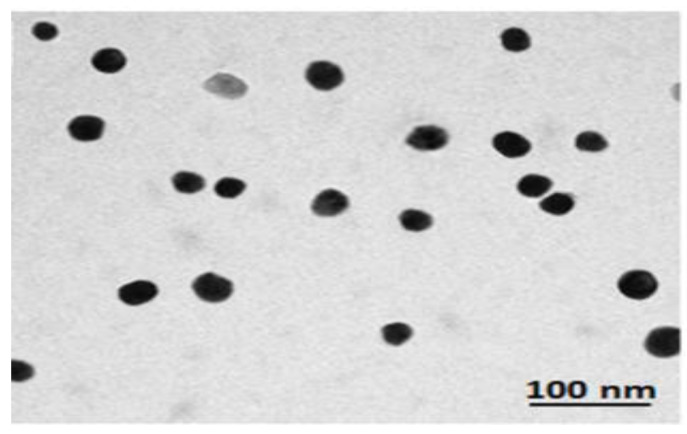
NSe morphology.

**Figure 3 biology-13-00253-f003:**
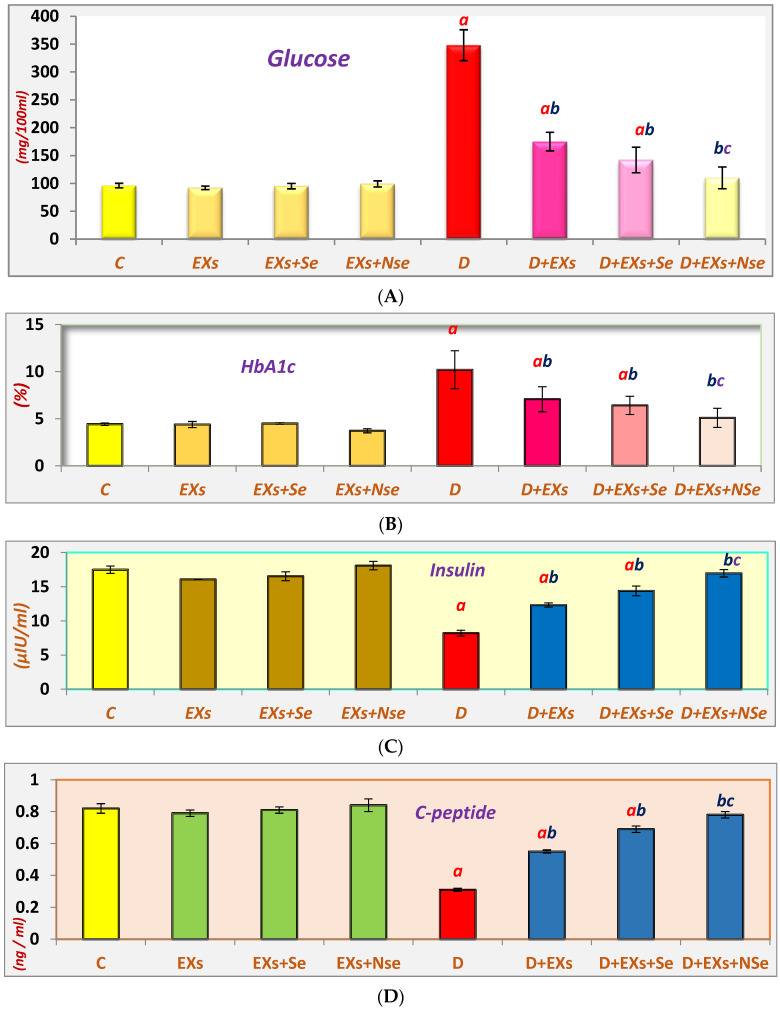
Values expressed as mean ± SEM (*n* = 8). (**A**) Serum glucose levels. (**B**) Serum HbA1c levels. (**C**) Serum insulin levels. (**D**) Serum C-peptide levels. Letters a, b, and c indicate significant differences (*p* ≤ 0.05) compared to control, untreated diabetic, and Ex+Se-treated diabetic groups, respectively.

**Figure 4 biology-13-00253-f004:**
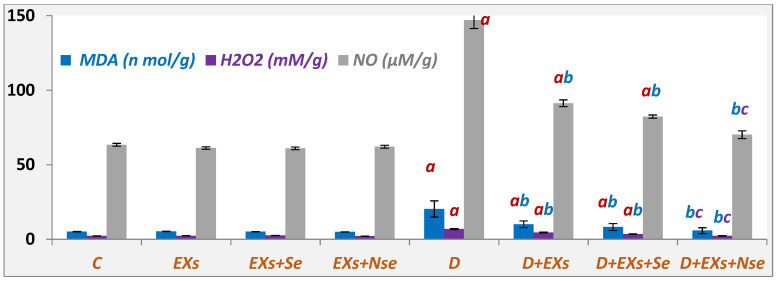
Pancreatic MDA, NO, and H_2_O_2_ contents. Values expressed as mean ± SEM (*n* = 8). Letters a, b, and c indicate significant differences (*p* ≤ 0.05) compared to control, untreated diabetic, and Ex+Se-treated diabetic groups, respectively.

**Figure 5 biology-13-00253-f005:**
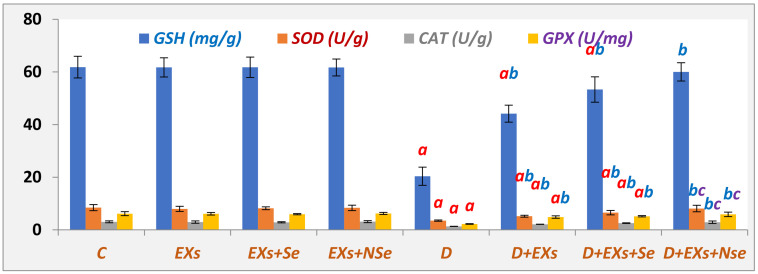
Pancreatic GSH, SOD, CAT, and GPx contents. Values expressed as mean ± SEM (*n* = 8). Letters a, b, and c indicate significant differences (*p* ≤ 0.05) compared to control, untreated diabetic, and Exs+Se-treated diabetic groups, respectively.

**Figure 6 biology-13-00253-f006:**
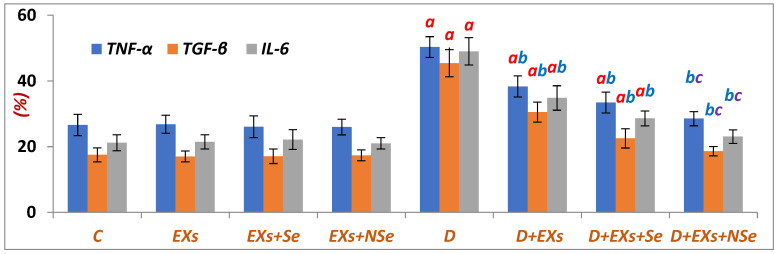
Pancreatic TNF-α, TGF-β, and IL-6 contents. Values expressed as mean ± SEM (*n* = 8). Letters a, b, and c indicate significant differences (*p* ≤ 0.05) compared to control, untreated diabetic, and Ex+Se-treated diabetic groups, respectively.

**Figure 7 biology-13-00253-f007:**
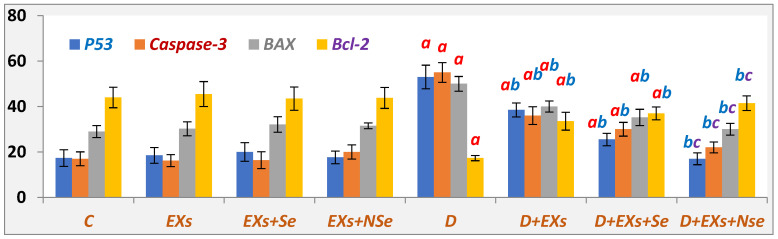
Pancreatic P53, BAX, Caspase-3, and Bcl-2%. Values expressed as mean ± SEM (*n* = 8). Letters a, b, and c indicate significant differences (*p* ≤ 0.05) compared to control, untreated diabetic, and *Ex+Se*-treated diabetic groups, respectively.

**Figure 8 biology-13-00253-f008:**
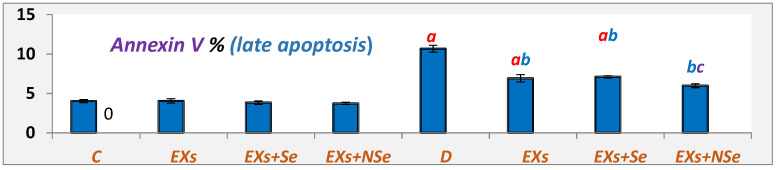
Pancreatic Annexin V (late apoptosis) %. Values expressed as mean ± SEM (*n* = 8). Letters a, b, and c indicate significant differences (*p* ≤ 0.05) compared to control, untreated diabetic, and EX+Se-treated diabetic groups, respectively.

**Figure 9 biology-13-00253-f009:**
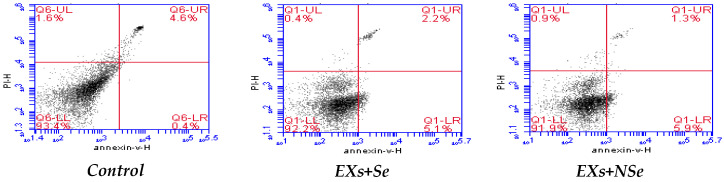
Pancreatic Annexin V % indications. **LL** = −ve for both stains = viable cells %. **LR** = +ve for annexin V = early apoptosis %. **UR** = +ve for both stains = late apoptosis %. **UL** = +ve for propidium iodide (PI) = necrosis %.

**Figure 10 biology-13-00253-f010:**
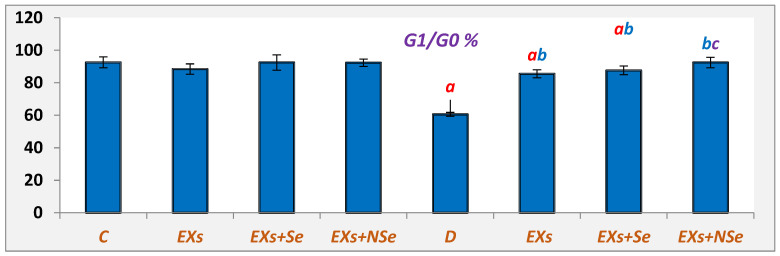
Pancreatic G1/G0%. Values expressed as mean ± SEM (*n* = 8). Letters a, b, and c indicate significant differences (*p* ≤ 0.05) compared to control, untreated diabetic, and Ex+Se-treated diabetic groups, respectively.

**Figure 11 biology-13-00253-f011:**
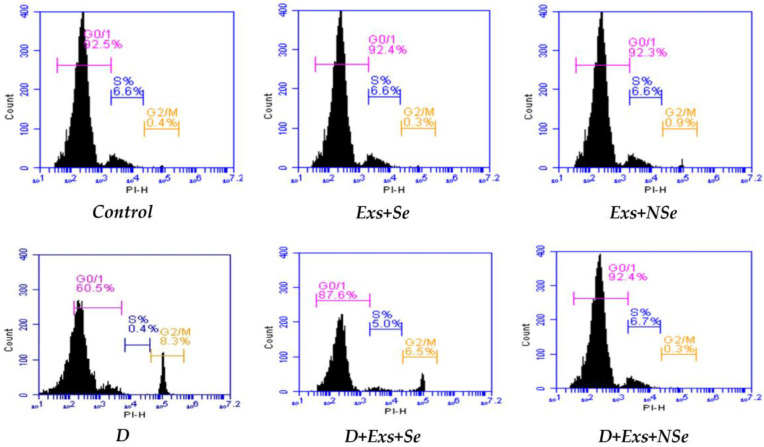
Pancreatic cell cycle (viable cells = G0/G1%).

**Figure 12 biology-13-00253-f012:**
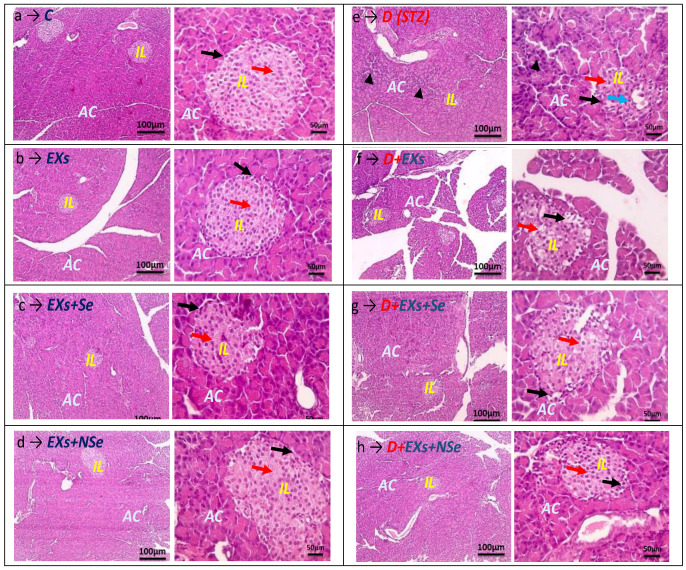
Photomicrograph of HE-stained pancreatic sections in different rat groups. (Low magnification → 100 X: bar 100 and high magnification → 400 X: bar 50). (**a**–**d**) *Control**,** Ex**,** Ex+Se* and *Ex+NSe* normal groups, respectively. These groups showed no histopathological alteration in the normal structure of either the exocrine portion, represented by acinar cells (***AC***), or the endocrine portion, represented by islands of Langerhans (***IL***), containing α-cells (black arrows) and β-cells (red arrows). (**e**) *D* (*STZ-treated*) group displayed evident ***IL*** degeneration, as seen in the severe hydropic degeneration of most *α-cells and β-cells*, with few *apoptotic β-cells* (blue arrow), and *pyknotic nuclei of epithelial lining acini* (arrowheads). (**f**–**h**) *Ex-, Ex+Se-*, and *Exs+NSe*-treated diabetic groups, respectively, demonstrated much less hydropic degeneration. However, pancreatic sections from the *D+Ex+NSe* group exhibited the most markedly improved histological picture of both *AC* and *IL (α* and *β-cells).*

## Data Availability

Data are contained within the article.

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
