# Peer review of "Mesenchymal Stem Cell-Derived Exosomes Loaded with Selenium or Nano Selenium as a Novel Therapeutic Paradigm for Streptozotocin-Induced Type 1 Diabetes in Rats"

_biology, 2024, doi:10.3390/biology13040253_

Round 1

Reviewer 1 Report

Comments and Suggestions for Authors

This manuscript is about the application of “mesenchymal stem cells-derived exosomes loaded with either selenium or nano selenium” in type 1 diabetes mellitus diseases. The manuscript can extend our knowledge in the field, provide an overview of current research about EXs-loaded with Se and EXs-loaded with NSe's role in T1DM diseases, and highlight how they can be used to develop novel therapies in T1DM diseases.

Overall, I thoroughly enjoyed reviewing it, but I have several requests for revision.

The authors address the following comments:

1. In the title it needs to imply, the model used.
2. Please change keywords according to Mesh terms and rearrange them according to the English alphabet.

3. The manuscript needs to be cleaned up for some punctuation and grammatical problems in the text.

4. Please use the abbreviation GPx for glutathione peroxidase.

5. Please turn IV or IP to italic.

6. What on-base choice is one dose, and what on-base choice is 0.5 ml? Please clarify.

7. In apoptosis analysis by flow cytometry it is mentioned by Annexin V but in Figure 3. I found the authors use Annexin V/Propidium Iodide (PI), please clarify it!

8. In methods authors mentioned frizzed tissue used for flow cytometry apoptosis (Annexin V/PI), as I know it needs fresh tissue. How do authors handle this?

9. Dear authors, it needs to be clarified why don’t statistically analyze EXs + Se and EXs + NSe with the other 3 groups.

10. The discussion section needs to be written properly in line with the manuscript analysis.

Comments on the Quality of English Language

The manuscript needs to be cleaned up for some punctuation and grammatical problems in the text.

Author Response

Comment 1: In the title it needs to imply, the model used.

Answer: Done as required.

Comment 2: Please change keywords according to Mesh terms and rearrange them according to the English alphabet.

Answer:  Done as required.

Comment 3: The manuscript needs to be cleaned up for some punctuation and grammatical problems in the text.

Answer:  Done as required.

Comment 4: Please use the abbreviation GPx for glutathione peroxidase.

Answer: Done as required.

Comment 5: Please turn IV or IP to italic.

Answer: Done as required.

Comment 6: What on-base choice is one dose, and what on-base choice is 0.5 ml? Please clarify.

Answer: Because we are planning shortly based on this study results to apply more than one dose. Using 0.5 ml was because we found it an appropriate amount rat can receive in the tail vein according to many previous studies with similar protocols. We would be pleased to take any suggestion regarding the dose in future studies.

Comment 7: In apoptosis analysis by flow cytometry it is mentioned by Annexin V but in Figure 3. I found the authors use Annexin V/Propidium Iodide (PI), please clarify it!

Answer:  PI is excited by wavelengths between 400 and 600 nm and emits light between 600 and 700 nm, and is therefore compatible with lasers and photodetectors commonly available in flow cytometers. This protocol for PI staining can be used to quantitate cell death in most modern research facilities and universities. Please see below reference.

Crowley LC, Scott AP, Marfell BJ, Boughaba JA, Chojnowski G, Waterhouse NJ. Measuring Cell Death by Propidium Iodide Uptake and Flow Cytometry. Cold Spring Harb Protoc. 2016 Jul 1;2016(7). doi: 10.1101/pdb.prot087163. PMID: 27371595.

Comment 8: In methods, authors mentioned frizzed tissue used for flow cytometry apoptosis (Annexin V/PI), as I know it needs fresh tissue. How do authors handle this?

Answer: Your point of view is right and we did use fresh specimens for flow cytometry. It was a writing mistake which is corrected in the text.

Comment 9: Dear authors, it needs to be clarified why don’t statistically analyze EXs + Se and EXs + NSe with the other 3 groups.

Answer: Because the other 3 groups had no significance with the control group. So their analysis with both treated diabetic groups (EXs + Se & EXs + NSe) would be the same as the control group.

Comment 10: The discussion section needs to be written properly in line with the manuscript analysis.

Answer: Done as possible.

Reviewer 2 Report

Comments and Suggestions for Authors

Author Response

Comment 1: Confirmation/Characterization/ Identification of percentage of the exosomes loaded with se/Nse are missing. Which is very important for the study to understand the exact dose of the Exosomes loaded with se/Nse used for the treatment. Usually this can be confirmed using Dynamic Light Scattering Technique (DLS). Without this data it is unpredictable to say the effect of se/Nse or Exosomes alone or with Exosomes+Se/Nse.

Answer:  You have shed light on an important point of course which will be taken into consideration in future design.

Comment 2: The Experimental design is not very clear to support the study. There is a need of a control group injected with “Exosomes alone” to delineate/to answer some of the concerns like the increased no of cells in early apoptosis phase compared to the control group (Table 6). 

Answer: We had already designed an exosome control group and another EXs-diabetic treated group which is under consideration for publication in a separate study in another journal for a while. So they were not included in this study to avoid duplication. However, our results reported no significant change between the exosome group and the control group. While EXs-diabetic treated rats showed significant improvements compared to the diabetic untreated rats, with less excellence than rats treated with EXs-loaded Se or NSe. A supplementary file with the non-included data was provided.

Comment 3: Data needs to be represented in a graphical format which will be easy to understand or to make clear conclusions (Table 1-6).

Answer: Such notice will be taken into consideration in future studies since currently, we have no one in our team who has the required experience to do so. If you could please accept our apology.

Comment 4: In line 72: Author mentioned that “In this line…” correct it as in this paper/ study.

Answer: The sentence was readjusted as required.

Comment 5: Authors highlighted some of the text in the manuscript example: line 68- 2018, line 71-insulin-signaling. What is the significance of these??

Answer: The highlight was not intentional

Comment 6: There are some grammatical and typographic errors are present.

Answer: Corrected as possible

Reviewer 3 Report

Comments and Suggestions for Authors

The current research contribution by Khalil is centered on the preparation and application of mesenchymal stem cells-derived exosomes loaded with selenium salt and selenium nanoparticle towards the treatment of type 1 diabetes. Type 1 diabetes is an intractable condition and this work presents a refreshing perspective on how to approach potential therapeutic development. The authors are to be commended for making the text accessible and easy for comprehension. A point that must equally be underscored is the fact that a number of issues arise from this work that ought to be addressed. Most notable is the lack of controls. Without any idea of the concentrations of the selenium and nanoselenium successfully entrapped in the exosomes, most of the data here become questionable. Also, no standard drug (API) used for treatment of type 1 diabetes was applied as a positive control, thus; there was no strong bases of comparison for the therapeutic efficacy. Other minor concerns are listed below.

-The authors are requested to include reported limitations and safety concerns associated with therapeutic application of selenium and selenium nanoparticles in the Introduction section.

-What specific gap is this research work attempting to fill.

-Lines 109-112: The washing time (30 min) was too short. How did the authors confirm that the washing was effective, i.e., no free (unentrapped) Se or NSe was left in the dialysis bag along with the exosomes?

-What was the concentration of selenium and NSe in the administered EX-loaded samples?

-Proper characterization of NSe is required. Authors stated that the size of NSe was 5-50 nm. This should be confirmed by DLS and measurement of the particle size from TEM. What was the nanoparticle surface charge and polydispersity index? Please perform XRD to better inform the reader pertaining to the crystallinity of NSe.

-With respect to the selenium used, what was the actual selenium salt purchased? Was it inorganic or organic, selenite or selenate. Please provide the full details.

Figure 1A, image of exosome morphology is too poor and unacceptable. First images of higher resolution and magnification should be provided. Secondly, authors should include TEM images of the selenium-loaded, nanoselenium-loaded and free-exosome.

-It appears from the TEM image in Figure 1A that a good number of the Se nanoparticles were not entrapped. Was that the case?

-The major challenge in in assessing this work has to do with a lack of necessary controls. First, the was no report on the concentration of the selenium or nanoselenium that was entrapped in the exosomes. Also, there was no positive treatment control group in which a typical diabetic drug was used. Finally, there was no variation of the drug agent in the treatment raising questions regarding the safety limit as well as the concentration or amount which is effective. All these issues have to be addressed. Authors should consider including this suggestions in a re-run of the animal studies.

-Selenium and to a lesser extent nano-selenium has a very narrow margin of safety as a micronutrient. Exceeding that limit, raises serious concerns about toxicity. Was there any preliminary studies performed. How was the issue of safety addressed?

The conclusions section is full of exaggerations and does not comport with the reality of the findings. It should be toned down. Authors should avoid hyperbolism. 

Comments on the Quality of English Language

The manuscript should be edited for English language. Importantly, the authors must also format the manuscript and reference in accordance with the author guideline for this journal.

Author Response

Comment 1: Most notable is the lack of controls. Without any idea of the concentrations of the selenium and nanoselenium successfully entrapped in the exosomes, most of the data here become questionable. 

Answer:  We had already designed an exosome control group and another EXs-diabetic treated group which is under consideration for publication in a separate study in another journal for a while. So they were not included in this study to avoid duplication. However, our results reported no significant change between the exosome group and the control group. While EXs-diabetic treated rats showed significant improvements compared to the diabetic untreated rats, with less excellence than rats treated with EXs-loaded Se or NSe. We will provide a supplementary file with the non-included data.

Comment 2: The authors are requested to include reported limitations and safety concerns associated with therapeutic application of selenium and selenium nanoparticles in the Introduction section.

Answer: Done as required.

Comment 3: What specific gap is this research work attempting to fill?

Answer: This work offers a new treatment protocol for T1DM which could be more effective when compared to previous agents.

 Comment 4: Lines 109-112: The washing time (30 min) was too short. How did the authors confirm that the washing was effective, i.e., no free (unentrapped) Se or NSe was left in the dialysis bag along with the exosomes?

Answer:  The washing time was 60 min (1 hour), not 30 and this was corrected in the revised version.

Comment 5: What was the concentration of selenium and NSe in the administered EX-loaded samples?

Answer:  We could not tell exactly since 20 ml of Se or NSe solutions were added to 20 ml of exosomes solution forming a Drug-EXs mixture, where obtained sample stocks were divided into insulin syringes (each of 0.5 mL). However, calculating concentration in the administered EX-load samples will be taken into consideration in future work regarding applying this protocol for addressing other tissue-diabetic complications.

Comment 6: Proper characterization of NSe is required. Authors stated that the size of NSe was 5-50 nm. This should be confirmed by DLS and measurement of the particle size from TEM. What was the nanoparticle surface charge and polydispersity index? Please perform XRD to better inform the reader pertaining to the crystallinity of NSe.

Answer: All these notes will be taken into consideration in future design since we purchased the NSe from Sigma Aldrich Co. (USA) who provided us with the outcomes of the characterization technique since it is not our point of specialty.

Comment 7: With respect to the selenium used, what was the actual selenium salt purchased? Was it inorganic or organic, selenite or selenate. Please provide the full details.

Answer:  We used elemental selenium.

Comment 8: Figure 1A, image of exosome morphology is too poor and unacceptable. First images of higher resolution and magnification should be provided. Secondly, authors should include TEM images of the selenium-loaded, nanoselenium-loaded and free-exosome.

Answer: Unfortunately, this was the only image provided by the supplier.

Comment 9: It appears from the TEM image in Figure 1A that a good number of the Se nanoparticles were not entrapped. Was that the case?

Answer:  Figure 1 A represents the morphological identification for the unloaded exosomes. It does not include any Se particles.

Comment 10: The major challenge in assessing this work has to do with a lack of necessary controls. First, the was no report on the concentration of the selenium or nanoselenium that was entrapped in the exosomes. Also, there was no positive treatment control group in which a typical diabetic drug was used. Finally, there was no variation of the drug agent in the treatment raising questions regarding the safety limit as well as the concentration or amount which is effective. All these issues have to be addressed. Authors should consider including these suggestions in a re-run of the animal studies.

Answer:  All such remarkable concerns will be taken into consideration in future work of course.

Comment 11: Selenium and to a lesser extent nano-selenium has a very narrow margin of safety as a micronutrient. Exceeding that limit, raises serious concerns about toxicity. Were there any preliminary studies performed? How was the issue of safety addressed?

Answer: We followed the loading protocol provided by the supplier which included using EX and either Se or NSe in 1:1 ratio, which finally make the concentration of Se or NSe inside EXs after loading is acceptable with the safe limit. Of course, assessing the exact loaded amount will be addressed in future design.

Comment 12: The conclusions section is full of exaggerations and does not comport with the reality of the findings. It should be toned down. Authors should avoid hyperbolism. 

Answer: Corrected as possible.

Round 2

Reviewer 2 Report

Comments and Suggestions for Authors

Comments are not addressed with scientific explanations.

Author Response

Following your recommendation, we have withdrawn the other article submitted to another journal containing the control and diabetic-treatment results regarding unloaded exosomes, and we merged all results in the current article provided with graphical results upon your request to make it easier for you to compare data and reach a conclusion. Unfortunately, the other analysis you requested cannot be carried out at this stage because no samples are left to do such an analysis. However, your valuable comments will be addressed in the future design indeed. Please accept our apology.

Reviewer 3 Report

Comments and Suggestions for Authors

The authors have made some marginal changes to the manuscript. However, the more substantive issues raised remained to be address. The claims made in the manuscript is not supported by sufficient data. It is understandable that some of the data had been included in another separate under consideration for publication in a different journal. But this practice is not acceptable. Thus, in its current form the manuscript cannot be recommeded for publication in Biology.

Comments on the Quality of English Language

The manuscript could benefit from professional English language editing.

Author Response

Following your recommendation, we have withdrawn the other article submitted to another journal containing the control and diabetic-treatment results regarding unloaded exosomes, and we merged all results in the current article provided with graphical results to make it easier for you to compare data and reach a conclusion. Unfortunately, the other analysis you requested cannot be carried out at this stage because no samples are left to do such an analysis. However, your valuable comments will be addressed in the future design indeed. Please accept our apology.

Round 3

Reviewer 2 Report

Comments and Suggestions for Authors

Author Addressed all the comments mentioned in the previous report. 

Author Response

Thank you for your great effort in reviewing our work.

Reviewer 3 Report

Comments and Suggestions for Authors

Some meaningful changes have been made to improve the quality of the manuscript. Nonetheless, it is still imperative for the readers to be provided with an objective view regarding the outcomes and more importantly, the limitations of this work. In view of this, authors should consider the following recommendations.

1. It should be made explicit as part of the limitations in the design of this work that a positive treatment control group i.e., diabetic rats treated with a standard anti-diabetic drug such as insulin or metformin was not included in this study. Other limitations such as not knowing the actual quantity of Nse that was successfully entrapped/loaded into the exosomes (meaning detailed knowledge of the active ingredient), and robust physicochemical characterization must be stated. Finally, authors should also make clear that the therapeutic limits of safety, therapeutic efficacy and toxicity of the drug was not ascertained since only 'one volume, 0.5 mL' was tested.

2. The Conclusions section must be toned-down. Loaded words such as 'excellences' should not be used, especially considering the backdrop of the point made in comment No. 1.

3. Professional English editing of the entire manuscript is required. There were so many instances of poor use of syntax which obfuscate the intended meaning of sentences. In its current form, the language of this text does not meet the academic standard of any quality journal.

Comments on the Quality of English Language

Extensive editing of English language is required.

Author Response

  1. The mentioned limitations were stated clearly at the end of the conclusion section.  
  2. The Conclusion section was reviewed as required.
  3. English editing was carried out as possible as we could.